# Nearest Neighbor Speculative Decoding for LLM Generation and Attribution

**Minghan Li**[1*], **Xilun Chen**[2], **Ari Holtzman**[3], **Beidi Chen**[2,4]
**Jimmy Lin**[5], **Wen-tau Yih**[2], **Xi Victoria Lin**[2]
[1] Cohere [2] Meta FAIR [3] University of Chicago
[4] Carnegie Mellon University [5] University of Waterloo
minghan@cohere.com, aholtzman@uchicago.edu, beidic@andrew.cmu.edu
jimmylin@uwaterloo.ca, {xilun, scottyih, victorialin}@meta.com

## Abstract

Large language models (LLMs) often hallucinate and lack the ability to provide attribution for their generations. Semi-parametric LMs, such as $k$NN-LM, approach these limitations by refining the output of an LM for a given prompt using its nearest neighbor matches in a non-parametric data store. However, these models often exhibit slow inference speeds and produce non-fluent texts. In this paper, we introduce **Ne**arest Neighbor **S**pecula**t**ive Decoding (NEST), a novel semi-parametric language modeling approach that is capable of incorporating real-world text spans of arbitrary length into the LM generations and providing attribution to their sources. NEST performs token-level retrieval at each inference step to compute a semi-parametric mixture distribution and identify promising span continuations in a corpus. It then uses an approximate speculative decoding procedure that accepts a prefix of the retrieved span or generates a new token. NEST significantly enhances the generation quality and attribution rate of the base LM across a variety of knowledge-intensive tasks, surpassing the conventional $k$NN-LM method and performing competitively with in-context retrieval augmentation. In addition, NEST substantially improves the generation speed, achieving a $1.8\times$ speedup in inference time when applied to Llama-2-Chat 70B. Code will be released at https://github.com/facebookresearch/NEST/tree/main.

## 1 Introduction

Large language models (LLMs) have demonstrated strong potential as multi-task solvers, excelling in a wide range of applications (Brown et al., 2020; Chowdhery et al., 2022; Touvron et al., 2023a; Anil et al., 2024). Despite their advanced capabilities, LLMs frequently encounter the problem of hallucination, particularly when dealing with long-tail knowledge that is less represented in their training data (Kandpal et al., 2023; Asai et al., 2023a). To address this limitation, retrieval augmentation incorporates information retrieval and nearest neighbour search from a non-parametric data store to enhance evidence-based and situated reasoning with LLMs. The resulting semi-parametric LMs exhibit a reduced tendency to generate unsupported content (Khandelwal et al., 2020; Borgeaud et al., 2022; Shi et al., 2024a,b; Asai et al., 2023a).

However, the effectiveness of retrieval-augmented language models (RALMs) in ensuring *accurate and reliable content generation* varies. The widely used in-context retrieval-augmentation (RA) regime (Ram et al., 2023; Shi et al., 2024a,b) softly biases the LM output distribution by prepending retrieved content to the input, which does not reliably guarantee faithful attribution of information. Approaches such as $k$NN-LM (Khandelwal et al., 2020) modify the LM output with a non-parametric

---

[*]Work done during internship at Meta.

38th Conference on Neural Information Processing Systems (NeurIPS 2024).

token distribution derived from nearest-neighbor matches in a corpus, which provide more direct attribution but has also been shown to degrade the quality of text generation (Wang et al., 2023a). Additionally, retrieval augmentation can significantly increase the *generation latency* due to the time required for the retrieval processes to complete and the subsequent expansion of the LM's context.

In this work, we propose **Ne**arest Neighbor **S**peculat**i**ve Decoding (NEST). This new semi-parametric language modeling approach is capable of incorporating real-world text spans of arbitrary length into the generations of an off-the-shelf LM, leading to improved quality and latency. NEST extends the standard $k$NN-LM approach, which interpolates the output distribution of an LM using the distribution of possible next tokens retrieved from a corpus (Khandelwal et al., 2020). It conducts an additional passage retrieval step at the beginning to limit the need to store and search over all tokens in the corpus, offering a balanced trade-off between search accuracy and efficiency. At each inference step, NEST performs content generation with three sub-steps:

**1) Confidence-based interpolation.** We use a novel *Relative Retrieval Confidence* (RRC) score to measure the uncertainty of the token retriever and use it as the interpolation coefficient of the output probability mixture. This enables flexible adaptation of the LM's output to different downstream tasks through dynamic interpolation with the token retrieval results.

**2) Dynamic span selection.** Inspired by the Copy Generator (COG) (Lan et al., 2023), NEST selects not only the best token predicted by the mixture probability but also extends to the span continuing from that token in the corpus when the token retrieval confidence exceeds a predefined threshold.

**3) Relaxed speculative decoding.** If a span of more than one token is selected, it undergoes evaluation based on the mixture probability. Through a rejection procedure similar to that in speculative decoding (Leviathan et al., 2023), only a prefix deemed highly likely by the mixture probability is accepted.

Evaluated on various free-form generation tasks—including question answering, text completion, and factuality-aware generation—using Llama-2-Chat models (Touvron et al., 2023b) of different sizes, NEST demonstrates superior performance compared to both the base LM and the standard $k$NN-LM under a zero-shot setting. For example, combined with NEST, the Llama-2-Chat 70B model demonstrates 42.3% improvement of ROUGE-1 on WikiText-103 and 21.6% improvement of FACTSCORE on Biography. Furthermore, NEST performs competitively with respect to in-context retrieval-augmentation on MMLU, Pile-of-Law, and TruthfulQA. We further demonstrate that the two approaches can be combined to enhance generation quality and attribution. Additionally, by generating multiple tokens at each time step, NEST significantly improves the efficiency of long-form generation. For Llama-2-Chat 70B, it achieves a $1.8\times$ speedup in inference time without compromising attribution or fluency.

## 2 Background

### 2.1 Problem Definition

Given an input $x$, a mixture model $\mathcal{M}$ predicts the output $y$ consisting of segments $\{y_1, y_2, ..., y_T\}$. In our setting, $\mathcal{M}$ may produce multiple tokens at a time step $t$, and therefore $y_t$ indicates the $t$-th segment consisting of at most $n$ tokens where $1 \leq |y_t| \leq n$. Let $\{w_t^{(1)}, w_t^{(2)}, ..., w_t^{(n)}\}$ be the tokens in segment $y_t$, we use $p_{\mathcal{M}}(w|x, y_{<t})$ to denote the distribution of the next token, and use $p_{\mathcal{M}}(w = w_t^{(1)}|x, y_{<t})$ to denote the probability of $w_t^{(1)}$ of the next segment $y_t$.

### 2.2 Nearest Neighbor Language Models ($k$NN-LM)

The mixture model $\mathcal{M}$ involves a pre-trained LM and key-value datastore $(\mathcal{K}, \mathcal{V})$ that enables approximate nearest neighbors search without further training or fine-tuning.

**Key-value datastore.** To create the datastore $(\mathcal{K}, \mathcal{V})$ using the LM for a corpus $\mathcal{D}$, let $f(\cdot)$ be the mapping from input sequence $c$ to the hidden states $h$ of the LM at some fixed layer. Let $w$ be the next word of $c$ in the corpus $\mathcal{D}$. For a sample $(c_i, w_i)$ in $\mathcal{D}$ after segmentation, we define the $i$-th key-value pair $(k_i, v_i)$ in $(\mathcal{K}, \mathcal{V})$ as $(h_i, w_i)$, where $h_i = f(c_i)$. The whole datastore is thus defined

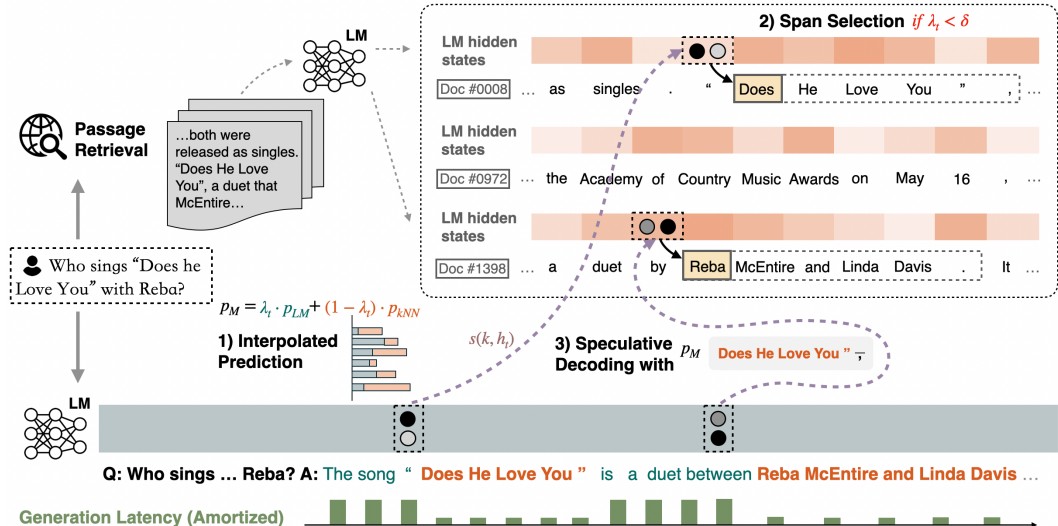

Figure 1: The NEST approach first locates the tokens in the corpus using the LM hidden states. The retrieval distribution $p_{k\text{-NN}}$ is dynamically interpolated with $p_{\text{LM}}$ based on the retriever's uncertainty $\lambda_t$. The token and its $n$-gram continuation are then selected from the mixture distribution $p_{\mathcal{M}}$, while the final span length is determined by speculative decoding to remove undesired tokens. The spans incorporated in the final generation provide direct attribution and amortize the generation latency.

as the set of all possible key-value pairs in $\mathcal{D}$:

$$(\mathcal{K}, \mathcal{V}) = \{(h_i, w_i) | (c_i, w_i) \in \mathcal{D}\}. \tag{1}$$

The size of the datastore $(\mathcal{K}, \mathcal{V})$ is proportional to the total number of tokens in corpus $\mathcal{D}$. This brings difficulty in scaling the size of the corpus and the model, which may require massive storage space and computational resources.

**Probability interpolation.** During inference, the language model outputs the token distribution $p_{\text{LM}}(w|x, y_{<t})$, together with the hidden state $q_t$. The model uses $q_t$ as a query to search the datastore $(\mathcal{K}, \mathcal{V})$ and retrieve the $r$-nearest neighbors $\pi$ according to the similarity $s(q, k)$ between a query $q$ and a key $k$. The final non-parametric distribution $p_{k\text{-NN}}(w|x, y_{<t})$ is computed using a softmax function over the similarity of all retrieved neighbors:

$$p_{k\text{-NN}}(w|x, y_{<t}) \propto \sum_{(k_i, v_i) \in \pi} \mathbb{I}_{w=v_i} \exp(\mu \cdot s(q_t, k_i)), \tag{2}$$

where $\mu$ is the inverse temperature. We use $1/\sqrt{\dim(q_t)}$ for $\mu$ in practice where $\dim(q_t)$ is the hidden state dimension. This is similar to computing attention in the Transformer model (Vaswani et al., 2017). For similarity $s(q, k)$, we follow Khandelwal et al. (2020) and use the negative squared $\ell_2$ distance. Items not in $\pi$ are assigned with 0 probability based on the indicator function $\mathbb{I}_{w=v_i}$.

Finally, the next token is sampled from the mixture distribution $p_{\mathcal{M}}$ of the non-parametric distribution $p_{k\text{-NN}}$ and the parametric distribution $p_{\text{LM}}$ using a fixed hyper-parameter $\lambda \in [0, 1]$:

$$p_{\mathcal{M}}(w|x, y_{<t}) = \lambda \cdot p_{\text{LM}}(w|x, y_{<t}) + (1 - \lambda) \cdot p_{k\text{-NN}}(w|x, y_{<t}). \tag{3}$$

## 3 Nearest Neighbor Speculative Decoding

### 3.1 Two-Stage $k$-NN Search

As mentioned in Section 2.2, maintaining a token-level key-value store can be expensive in terms of both latency and storage. To provide a better trade-off between latency and accuracy, we adopt the two-stage design, which is widely applied in information retrieval and search engines.

**First-stage passage retrieval**    Given the corpus $\mathcal{D}$, we segment the documents into separate passages of less than $m$ tokens each. We then encode the corpus and use a hybrid retrieval system to select the relevant passages, as dense retrievers are good at handling semantics in queries (Karpukhin et al., 2020) and sparse retrievers are good at lexical matching (Sciavolino et al., 2021).

**Second-stage $k$-NN token search**    After obtaining the top-$b$ retrieved passages $\{d_1, d_2, ..., d_b\}$ at time step $t$, we use the encoder $f(\cdot)$ of LM to encode the prefixes of all tokens as keys as shown in Figure 1. The key-value datastore $(\mathcal{K}, \mathcal{V})$ therefore is created *on the fly*. Similarly, we use the negative squared $\ell_2$ distance as the similarity function and $q_t$ as the queries to search for the top-$r$ nearest neighbors $\pi$ in $(\mathcal{K}', \mathcal{V}')$.

The two-stage design provides a trade-off between search latency and accuracy and the passage-level index only takes a fraction of the token-level index in Section 2.2. In addition, the first-stage passage search also acts as a filter to remove deceptively similar tokens in non-relevant contexts.

## 3.2 Confidence-Based Output Interpolation

Similar to Equation (3), we linearly interpolate the language model's distribution $p_{\text{LM}}$ and non-parametric distribution $p_{k\text{-NN}}$ using a coefficient $\lambda_t$ for a time step $t$ in generation. The difference is that we use the token retrieval score to compute $\lambda_t$:

$$\lambda_t = \sigma\left(\left(\frac{\min_i |s(q_t, k_i)|}{\max_i |s(q_t, k_i)|} - \alpha\right)/\tau\right),$$
(4)

where $\sigma$ is the sigmoid function and the min-max ratio expresses the uncertainty of the $k$-NN component. We use the sigmoid activation to re-center and re-scale this uncertainty, where $\alpha$ is the offset and $\tau$ is the scale for the sigmoid function. We refer to this method as Relative Retrieval Confidence (RRC).

If the downstream task does not involve generation, such as perplexity evaluation and multi-choice tasks, our method will end at Equation (4). The mechanisms introduced in the following sections are only applied to generation, including token/span selection and post-hoc revision.

## 3.3 Dynamic Span Selection

Directly sampling tokens from the mixture distribution $p_{\mathcal{M}}$ might escalate the exposure bias since the seemingly coherent tokens might be retrieved from completely different sources. To maintain coherence, we extend the context of the current sampled token by using its $n$-gram continuation in the corpus. Given the current time step $t$, we first select the next token $w_t$ from the mixture distribution $p_{\mathcal{M}}$. However, the sampled token $w_t$ may correspond to multiple retrieved $w_i$ (i.e., the value $v_i$), in the neighbors $\pi$ which have different $n$-gram continuations. We use a simple max-pooling strategy[2] to select the starting token $w_t^{(1)}$ of the $n$-gram from $\pi$:

$$w_t^{(1)} = \underset{\{w_i | w_i = w_t, w_i \in \pi\}}{\operatorname{argmax}} p_{k\text{-NN}}(w = w_i | x, y_{<t})$$
(5)

The corresponding $n$-gram for time step $t$ is $\{w_t^{(1)}, w_t^{(2)}, ..., w_t^{(n)}\}$ where $n$ is fixed hyper-parameter. The final output is determined by the interpolation coefficient $\lambda_t$ in Equation (4):

$$y_t = \begin{cases} w_t, & \text{if } \lambda_t > \delta; \\ \{w_t^{(1)}, w_t^{(2)}, ..., w_t^{(n)}\}, & \text{otherwise.} \end{cases}$$
(6)

where $\delta$ is a threshold and $y_t$ is the segment output at time step $t$.

## 3.4 Relaxed Speculative Decoding

Despite the dynamic selection, the hyper-parameter $n$ is hard to control over different tasks. To produce spans with adaptive length, we take inspiration from Leviathan et al. (2023), where we use $\mathcal{M}$ to revise the proposed $n$-gram. However, the proposal distribution $q(w|x, y_{<t})$ is unknown

---

[2]We used a slightly different implementation to ensure the sampled token is in $\pi$. Please see the code here: https://github.com/facebookresearch/NEST/blob/main/models/knn_transformers.py

besides the first token $w_t^{(1)}$. Therefore, we use a relaxed version of speculative decoding that upper bounds the acceptance probability. The probability of accepting the token $w_t^{(i)}$ in a span is:

$$P(\text{accept token } w_t^{(i)}) = \min\left(1, \frac{p_{\mathcal{M}}(w = w_t^{(i)} \mid x, y_{<t}, w_t^{(1)}, w_t^{(2)}, ..., w_t^{(i-1)})}{\gamma \cdot \max_w p_{\mathcal{M}}(w \mid x, y_{<t}, w_t^{(1)}, w_t^{(2)}, ..., w_t^{(i-1)})}\right), \quad (7)$$

where $\gamma \in (0, 1]$ is the relaxation factor, which is referred to as "leniency" by Leviathan et al. (2023). The smaller $\gamma$ is, the less often $\mathcal{M}$ rejects the draft. If token $w_t^{(i)}$ is rejected, we will remove all the tokens from $w_t^{(i)}$ to $w_t^{(n)}$, and then re-select a token $w_t^{(i)}$ from the distribution $p_{\mathcal{M}}$ without going through the span selection. The computation for processing multiple tokens can be parallelized and NEST can thus maintain the latency or even accelerate the generation. Moreover, suppose all tokens in the draft are not rejected. In that case, we will directly fetch the $n$-gram's continuation in the corpus and use it for the next draft proposal until rejection, removing the reliance on the hyper-parameter $n$.

Once the $n$-gram is accepted, the corresponding parts are masked in the corpus and will never be used again in this generation. This is to prevent the $k$-NN component from repetitively retrieving the same segments in a small key-value store $(\mathcal{K}', \mathcal{V}')$. We provide the complete procedure in Algorithm 1.

## 4 Experiments

We evaluate NEST and other baselines on various tasks including text completion, question-answering, fact-verification, and multi-choice tasks, providing a comprehensive picture of factuality, fluency, and attribution of NEST in different domains. In all experiments, we focus on evaluating instruction-following models. We use Llama-2-chat under a zero-shot setting, where we remove the few-shot demonstrations from the instructions to simulate the realistic usage of these models.

### 4.1 Benchmark Datasets

**Text completion.** WikiText-103 (Merity et al., 2017) is a standard benchmark for language modeling, extracted from the set of verified articles on Wikipedia. **Pile of Law** (Henderson et al., 2022) is a growing dataset of legal and administrative data. We use the datasets[3] from Huggingface and further split the test data into validation and test sets. For language modeling, we report the perplexity score. For free-form generation, we report ROUGE-1, 2, L (Lin, 2004) and MAUVE (Pillutla et al., 2021).

**Question answering.** We select four knowledge-intensive question-answering datasets, including Natural Questions (NQ) (Kwiatkowski et al., 2019), TriviaQA (TQA) (Joshi et al., 2017), HotpotQA (HQA) (Yang et al., 2018), and MedMCQA (MQA) (Pal et al., 2022). Since the in-context demonstrations are removed for free-form generation, we use answer-level recall (i.e., Hit@1) (Karpukhin et al., 2020) which checks if the output contains any correct answers instead of exact match.

**Fact verification.** We evaluate a biography-generation task (Min et al., 2023) and Truthful-QA (Lin et al., 2022) which is a benchmark for testing false beliefs or misconceptions. We use FACTSCORE (Min et al., 2023) for biography. For TruthfulQA, we follow Lin et al. (2022) which uses the difference between the max similarity to a true reference answer and the max similarity to a false reference answer for BLEU and ROUGE-1.

**Closed-set tasks.** MMLU (Massive Multitask Language Understanding) (Hendrycks et al., 2021) benchmark covers 57 subjects across STEM, the humanities, the social sciences, and more. We report the macro accuracy for each domain.

### 4.2 Implementation

**Knowledge Sources.** Wikipedia (CC BY-SA 3.0): For tasks except text completion on Pile of Law, we use the Wikipedia 2021 dump released by Izacard et al. (2024) as the knowledge source and follow the same pre-processing procedures in RA-DIT (Lin et al., 2024), yielding $\sim$33M passages with each

---

[3] `https://huggingface.co/datasets/pile-of-law/pile-of-law/tree/main`

less than 200 tokens. Pile of Law (CC BY-NC-SA 4.0): We use the training split from Huggingface and select only the English data. We then follow the same procedure applied in Wikipedia, yielding a corpus containing ∼15M passages after filtering. More details are provided in Appendix A.

**Inference setting.** $k$NN-LM and NEST share the same first-stage retriever. We use DRAGON+ (Lin et al., 2023) and BM25 (Robertson and Zaragoza, 2009) to encode the segments into dense and sparse vectors, respectively. Given the input, we query both the dense and sparse indexes at the same time and retrieve their corresponding top-$(b \cdot l)$ passages. We linearly interpolate the similarity scores between the two search results (also known as fusion) and sort them before selecting the top-$b$ passages. The number of passage candidates $b$ is set to be 40 and the scaling factor $l$ is set to be 100. For RA, we use the top-3 passages in the prompt due to the context window limit. We further combine NEST and RA since they are independent methods. Greedy decoding is used during generation. More details about retrieval, decoding, and hyper-parameters are described in Appendix B.

**Evaluation setting.** For text completion tasks and perplexity evaluation, we use 128 tokens as the prefix and the consecutive 256 tokens as the target. For the other tasks, we use 128 tokens as the max generation length for question answering and 512 for fact verification. For retrieval-based models, only the prefix will be used for retrieval. Hyper-parameters of all baselines and NEST are tuned on the dev set of WikiText-103, NQ, and Biography. Each baseline uses the same hyper-parameters for all tasks evaluated. We first tune the related hyper-parameters for perplexity and then tune the rest for generation metrics to reduce the search space. More details are provided in Appendix B.

### 4.3 Baselines

**Base LMs.** We evaluate publicly available, instruction-tuned language models, Llama-2-chat series[4], with model sizes ranging from 7B, 13B to 70B.

**Two-Stage $k$NN-LM.** We apply the two-stage strategy described in Section 3.1 to $k$NN-LM as well, where we retrieve the top-$b$ passages and encode a key-value datastore $(\mathcal{K}', \mathcal{V}')$ on the fly.

**In-Context Retrieval Augmentation (RA).** A common retrieval-augmentation method is adding the retrieved evidence into the prompt. We perform retrieval given the only input instead of retrieving new passages every $k$ step due to the expense of refreshing the kv-cache.

### 4.4 Main Results

Table 1 shows the main results of NEST and other baselines. For **language modeling**, RA-NEST is able to achieve the lowest complexity on both WikiText-103 and Pile of Law. For **text completion**, RA has the best MAUVE scores and ROUGE scores in Wikitext-103 while RA-NEST works better for 7B and 13B models on Pile of Law. We observe that for legal documents, quoting the exact clauses from the source might be more favourable compared to Wikipedia.

For **question-answering**, RA-NEST tends to work better for smaller models (7B and 13B) in general. The gap between base LMs and other methods diminishes for 70B LMs, which is consistent with previous work where retrieval is found most useful for smaller models (Borgeaud et al., 2022).

For **fact-verification**, NEST is able to outperform the base LMs but underperform RA in terms of the FACTSCORE. RA-NEST is able to outperform RA for the 70B model. The degradation for RA-70B is caused by generating shorter claims which is punished by the FACTSCORE. On TruthfulQA, the semi-parametric LMs consistently outperform base LMs and RAs where in-context retrieval seems to have a negative effect on the scores. This is because TruthfulQA is an adversarial dataset containing difficult questions where in-context RA is more susceptible to the "evidence" in the prompt (e.g., astrology and myths). In contrast, NEST only interpolates the results at the output level and therefore performs better in this case. The combination RA-NEST is also affected by the in-context retrieval.

For **closed-set tasks**, NEST is comparable to RA and RA-NEST manages to achieve the best macro scores on average. Overall, NEST is able to outperform base LMs and $k$NN-LM's on most tasks while being on par with RA. The combination of RA and NEST further improves over the two methods

---

[4]https://huggingface.co/meta-llama/Llama-2-70b-chat-hf

| Models | Wikitext-103 | | | | | | Pile of Law | | | | | |
|---|---|---|---|---|---|---|---|---|---|---|---|---|
| | PPL($\downarrow$) | MAUVE | RG-1 | RG-2 | RG-L | Avg. Len | PPL($\downarrow$) | MAUVE | RG-1 | RG-2 | RG-L | Avg. Len |
| Llama-2-Chat$_{7B}$ | 14.6 | 58.8 | 15.8 | 3.7 | 14.4 | 175.4 | 10.1 | 80.7 | 19.1 | 5.5 | 17.1 | 211.4 |
| +RA | 7.2 | 74.6 | **35.7** | **23.1** | **34.4** | 204.5 | 7.1 | 84.7 | 23.1 | 8.9 | 21.1 | 222.0 |
| +$k$NN-LM | 9.8 | **82.5** | 23.7 | 7.7 | 21.7 | **238.2** | 8.8 | 81.1 | 19.4 | 5.7 | 17.4 | 214.3 |
| +NEST | 8.4 | 73.2 | 28.4 | 14.2 | 27.1 | 218.4 | 8.1 | 88.0 | 23.7 | 8.7 | 21.5 | 226.5 |
| +RA-NEST | **6.4** | 72.6 | 35.2 | 22.7 | 34.0 | 202.0 | **6.7** | **90.0** | **24.4** | **9.0** | **22.2** | **232.1** |
| Llama-2-Chat$_{13B}$ | 12.0 | 75.9 | 19.9 | 4.9 | 18.0 | 218.4 | 8.2 | 72.8 | 17.5 | 5.3 | 15.7 | 181.7 |
| +RA | 6.5 | **91.5** | **38.9** | **24.2** | **37.2** | **249.3** | 5.9 | 86.6 | 23.6 | 9.1 | 21.5 | 228.7 |
| +$k$NN-LM | 8.6 | 76.3 | 23.7 | 8.2 | 21.9 | 238.5 | 7.4 | 71.5 | 17.7 | 5.3 | 15.9 | 183.7 |
| +NEST | 7.2 | 67.1 | 29.3 | 15.6 | 28.1 | 207.1 | 6.8 | 86.0 | 22.9 | 8.7 | 20.9 | 212.3 |
| +RA-NEST | **5.8** | 86.8 | 38.6 | 24.0 | 37.0 | 245.5 | **5.7** | **90.1** | **24.7** | **9.2** | **22.4** | 229.4 |
| Llama-2-Chat$_{70B}$ | 9.9 | 88.6 | 22.9 | 6.2 | 20.8 | 239.6 | 6.9 | 93.4 | 23.0 | 7.1 | 20.7 | 250.1 |
| +RA | 5.3 | 91.6 | **40.5** | **26.1** | **38.8** | 235.9 | 4.9 | 95.5 | **26.3** | **10.1** | **24.0** | 253.2 |
| +$k$NN-LM | 7.1 | 83.6 | 26.1 | 9.6 | 24.1 | **253.9** | 6.3 | 94.4 | 23.1 | 7.2 | 20.8 | 251.3 |
| +NEST | 6.3 | 82.6 | 32.6 | 17.2 | 31.1 | 236.3 | 5.9 | 95.4 | 25.6 | 9.4 | 23.2 | 251.3 |
| +RA-NEST | **4.8** | 90.0 | 40.2 | 25.9 | 38.6 | 233.1 | **4.7** | **97.6** | 26.2 | 9.5 | 23.7 | **253.6** |

| Models | TQA | NQ | HQA | MQA | Avg. | TruthfulQA | | Biography | | MMLU | | | | |
|---|---|---|---|---|---|---|---|---|---|---|---|---|---|---|
| | | Answer-Level Recall | | | | $\Delta$BLEU | $\Delta$RG-1 | FS | # Facts | Human. | STEM | Social | Other | Avg. |
| Llama-2-Chat$_{7B}$ | 61.1 | 38.9 | 30.6 | 9.3 | 35.0 | -0.02 | 0.42 | 27.2 | **71.2** | 37.8 | 32.6 | 38.9 | 39.6 | 37.2 |
| +RA | **69.5** | 48.4 | 44.1 | 12.8 | 43.7 | -0.34 | 0.18 | **56.5** | 67.1 | 41.8 | 35.3 | **42.2** | 43.3 | 40.7 |
| +$k$NN-LM | 63.4 | 42.4 | 33.5 | 9.5 | 37.2 | **0.13** | **0.66** | 30.6 | 59.8 | 38.0 | 33.1 | 39.2 | 40.1 | 37.6 |
| +NEST | 61.5 | 43.2 | 33.5 | 10.2 | 37.1 | 0.03 | 0.45 | 38.9 | 58.2 | **42.0** | **35.4** | 42.0 | **43.4** | **40.7** |
| +RA-NEST | 69.0 | **48.8** | **45.3** | **13.3** | **44.1** | -0.32 | 0.21 | 55.1 | 57.7 | 37.9 | 32.7 | 39.3 | 39.8 | 37.4 |
| Llama-2-Chat$_{13B}$ | 63.5 | 42.3 | 32.6 | 10.2 | 37.2 | 0.13 | 0.81 | 28.8 | 49.9 | 41.5 | 35.0 | 40.2 | 43.8 | 40.1 |
| +RA | 70.9 | 51.6 | 44.6 | 14.0 | 45.3 | -0.16 | 0.25 | **59.1** | 51.2 | 43.4 | 37.4 | 43.5 | 46.4 | 42.7 |
| +$k$NN-LM | 64.7 | 43.5 | 34.2 | 11.2 | 38.4 | 0.20 | 0.95 | 31.1 | 46.1 | 41.4 | 34.7 | 40.6 | 44.2 | 40.2 |
| +NEST | 64.2 | 44.2 | 34.3 | 10.9 | 38.4 | **0.29** | **0.98** | 35.7 | 47.2 | 41.3 | 34.9 | 40.2 | 43.7 | 40.0 |
| +RA-NEST | **70.9** | **51.7** | **45.3** | **14.7** | **45.7** | -0.14 | 0.25 | 58.4 | 52.4 | **43.5** | **37.7** | **43.5** | **46.7** | **42.8** |
| Llama-2-Chat$_{70B}$ | 74.0 | 50.1 | 39.5 | 12.8 | 44.1 | 0.14 | 0.70 | 34.2 | **58.8** | 43.5 | 37.9 | 44.4 | 47.0 | 43.2 |
| +RA | **75.5** | **55.4** | **52.5** | 16.0 | **49.9** | -0.13 | 0.40 | 52.9 | 42.1 | **45.9** | 39.7 | 46.2 | 48.6 | 45.1 |
| +$k$NN-LM | 74.6 | 51.2 | 40.2 | 13.5 | 44.9 | 0.08 | 0.58 | 36.1 | 54.4 | 44.0 | 37.4 | 44.1 | 47.1 | 43.2 |
| +NEST | 74.2 | 51.6 | 41.4 | 13.8 | 45.2 | **0.17** | **0.70** | 41.6 | 56.2 | 43.8 | 38.0 | 44.4 | 47.8 | 43.5 |
| +RA-NEST | 75.4 | 55.2 | 52.4 | **16.3** | 49.8 | -0.19 | 0.31 | **59.2** | 53.8 | 45.8 | **39.7** | 46.2 | **48.9** | 45.1 |

Table 1: Results on text completion (upper table) and other tasks (lower table). Bold numbers indicate the best performance. PPL: Perplexity. RG: ROUGE score. Avg. Len: Average generation length. $\Delta$BLEU/$\Delta$RG: The difference between the max score to correct references and the max score to incorrect references. FS: FACTSCORE with length penalty.

on some tasks. Despite the limited improvement, we will show that NEST is able to provide better attribution and latency in the following sections.

## 4.5 Latency Analysis

**Latency breakdown.** The combination of dynamic span selection and relaxed speculative decoding can improve the latency of the LLM generation by quick draft proposal and processing multiple tokens at a time step. Figure 2a shows the latency breakdown of a NEST-70B model ($\alpha = 0.3, \tau = 0.1, \delta = 0.5$) for different relaxation factors on the Biography validation data. The latency experiment is done on 8×A100 GPUs (for model parallelization) and 32 CPU threads (for search). The batch size is set to 1. We use internal, research-purpose implementation of the base Llama-2-chat model which did not optimize for latency. As we can see, the LM encoding time takes about half of the latency, while the sum of the others takes the rest. Noticeably, the cost of passage search and token index building stay relatively constant per query, while the others are related to the number of tokens processed per time step. Still, even with extra retrieval overheads, the slowest NEST model is faster than the base LM, showing the efficacy of span selection and speculative decoding.

**Latency-accuracy trade-off.** To understand why NEST can accelerate generation, we first show the latency-accuracy trade-off by tuning the relaxation factor in Figure 2b. The smaller $\gamma$ is, the less often NEST rejects a segment retrieved from the corpus, which enables more tokens to be processed in parallel. The average proposed span length in Figure 2a can increase from 5 tokens to 35 tokens at each time step as the relaxation factor gets smaller. Combined with Figure 2a, we can reach the conclusion that fetching longer spans from the corpus results in lower generation latency per query. For the accuracy, the FACTSCORE on Biography validation data shows that there is a sweet spot around $\gamma = 5e - 2$ where both low latency and high accuracy can be achieved at the same time.

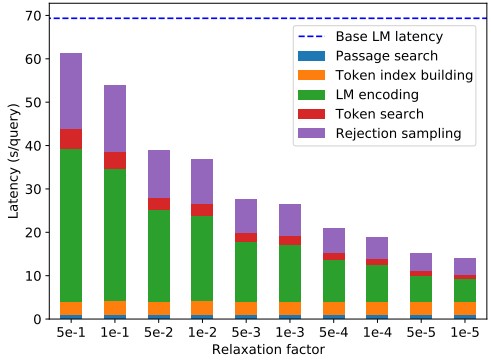

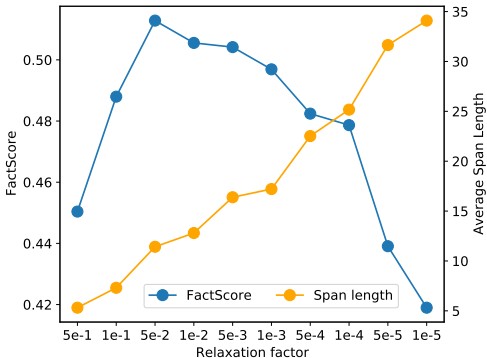

(a) Latency Breakdown.

(b) Span length and FACTSCORE on Biography.

Figure 2: Latency-accuracy trade-off and breakdown on Biography using Llama-2-Chat 70B+NEST. As the relaxation factor $\gamma$ decreases, NEST tends to accept longer spans from the corpus. We choose $\gamma = 5e-2$ in our main experiments, which accelerates the generation and improves the FACTSCORE.

| Task | Models | Ratio (%) | Length | Example (Generated by NEST-70B) |
|---|---|---|---|---|
| NQ | NEST 7B | 93.4 | 18.4 | Question: who sings does he love me with reba? |
| | NEST 13B | 52.4 | 4.4 | Answer: The song "Does He Love You" [1] is a duet |
| | NEST 70B | 58.8 | 4.3 | between Reba McEntire and Linda Davis.  It was |
| | RA-NEST 7B | 77.5 | 3.1 | released in August 1993 as the first single from Reba's |
| | RA-NEST 13B | 33.2 | 3.0 | compilation album Greatest Hits Volume 2. It is one of |
| | RA-NEST 70B | 44.9 | 3.9 | country music's several songs about a love triangle. [2] |
| Bio | | | | Question: Tell me a bio of Taral Hicks. |
| | | | | Answer: Taral Hicks (born September 21, 1974) is an American actress |
| | NEST 7B | 95.5 | 27.9 | and singer. Hicks is best known for her acting in such films as 1993's |
| | NEST 13B | 53.9 | 10.6 | American crime drama film A Bronx Tale and her singing in such work |
| | NEST 70B | 58.6 | 7.0 | as her 1997 [3] debut studio album This Time which peaked at No. 4 on |
| | RA-NEST 7B | 50.3 | 5.1 | Billboard Bubbling Under Hot 100 Singles. Raised in Teaneck, |
| | RA-NEST 13B | 48.5 | 5.9 | New Jersey, Hicks graduated from Teaneck High School in 1994 [3]. |
| | RA-NEST 70B | 80.7 | 11.0 | She is the younger sister of actress and singer D'atra Hicks. |

Table 2: Attribution analysis. (Attribution) Ratio: Proportion of tokens that are taken from the corpus. (Attribution) Length: Average length of consecutive spans in the generation that are taken from the same document. Green: Segments taken from the corpus. Gray: Reference.

## 4.6 Attribution and Qualitative Analysis

One of the most important features of NEST is providing attribution directly at a span level, where the reference for the corresponding statement is accurate since it is directly taken from the corpus. Table 2 shows the attribution ratio, average attributed span length, and two examples for analysis. For NQ and Biography tasks, depending on the model and hyper-parameters in Equation (4) and (7), the ratio of tokens that can be traced back to the corpus ranges from 33.2% to 95.5%. In addition, it is more desirable to have consecutive segments that come from the same source so that consistent attribution can be provided, and the average length of spans taken from the corpus ranges from 3.0 to 27.9 tokens. This feature provides span-level attribution for most claims in the LLM generation. To our knowledge, neither of the baselines can achieve the same granularity and preciseness for the attribution as NEST. We provide more analyses on sensitivity and ablation for NEST in Appendix C.

## 5 Related Work

### 5.1 Retrieval-Augmentation

Retrieval Augmentation involves external knowledge sources to improve the effectiveness of language models on knowledge-intensive tasks. Chen et al. (2017) propose DrQA which combines extractive models and independent retrievers for open-domain question-answering. Follow-up works on retrieval-augmentation such as REALM (Guu et al., 2020), RAG (Lewis et al., 2020), and Atlas (Izacard

et al., 2024) further combine the retrieval component in pre-training and fine-tuning for downstream knowledge-intensive tasks. Asai et al. (2024) further divide them into three categories:

**Input augmentation.** REPLUG (Shi et al., 2024a) and in-context RALM (Ram et al., 2023) propose to pre-pend the retrieved passages in the prompts for zero-shot factual support. Recently, Self-RAG (Asai et al., 2023b) leverages special tokens to perform adaptive retrieval and different critics to iterative refine the RALM's output. RA-DIT (Lin et al., 2024) retrofits LLMs with retrieval capabilities via instruction fine-tuning.

**Intermediate fusion.** RETRO (Borgeaud et al., 2022) employs a novel attention mechanism to incorporate multiple pre-processed text fragments in intermediate layers for more efficient integration of retrieved results. This approach has been successfully applied to larger decoder-only language models as demonstrated by RETRO++ (Wang et al., 2023b) and InstructRetro (Wang et al., 2024). FiD (Izacard and Grave, 2021) applies similar an encoder-decoder structure in a zero-shot manner and achieves better effectiveness at a document level.

**Output integration.** $k$NN-LM (Khandelwal et al., 2020) pioneers this direction and proposes to interpolate the retrieval distribution and LM's prediction. Follow-up works further propose adaptive interpolation methods which involve training (He et al., 2021; Bhardwaj et al., 2023) and excessive tuning (Drozdov et al., 2022). Another line of work proposes to joint train the phrase encoder and LM to expand the vocabulary dynamically using the retrieved phrases, such as Copy-Generator (Lan et al., 2023) and its follow-up work (Cao et al., 2024). Martins et al. (2022) proposes a chunk-based $k$NN machine translation model which retrieves chunks of tokens from the datastore.

## 5.2 Inference-Time Revision

Speculative decoding (Leviathan et al., 2023; Chen et al., 2023; Miao et al., 2023; Spector and Re, 2023) is an acceleration method that leverages a small model to generate drafts for a large model to evaluate. The latency is improved as the larger model can process multiple tokens in parallel at each time step. Recently, REST (He et al., 2024) proposes to draw multiple drafts from a datastore and leverages a prefix trie tree to compute the proposal distribution, which is the closest concurrent work. Yang et al. (2023) also utilizes prefix matching to select draft sentences from a datastore, and keep the continuation of the draft sentence as long as the token matches with the model generation.

In general, speculative decoding can be categorized as an unbiased self-revision method. In comparison, NEST changes the LM output distribution through interpolation with a non-parametric probability distribution. Previous work focusing on fact-checking follows a similar idea to generate factually consistent texts with a set of evidence via post-hoc editing, such as FRUIT (Iv et al., 2022) and PEER (Schick et al., 2022). Recently, RARR (Gao et al., 2023) leverages more complex planning with LLMs to verify the retrieved evidence and generate attribution reports.

## 6 Limitations

While being able to directly retrieve segments from the corpus and apply them in the generation, the output of NEST might still contain factual errors depending on the accuracy of the first-stage passage retrieval and the second-stage token retrieval. Moreover, as a plug-and-play method, our main goal is to provide a flexible solution that can combine different LLMs and data stores in zero- and few-shot manners. Without further fine-tuning, the integrated system might be sub-optimal and the results can be better if it is fine-tuned on appropriate tasks. Lastly, such semi-parametric LMs may not improve the ability of in-context learning, since the demonstrations in the prompts are unlikely to appear in any contexts that can be found in the database. An observation from preliminary experiments is that the current neural retrievers do not have the capability to process the in-context few-shot information, where techniques such as query reformulation might be needed for parsing the demonstrations.

## 7 Conclusion

This paper presents NEST, an inference-time revision method for LMs that improve their factuality and attribution through nearest neighbor speculative decoding. Leveraging two-stage $k$-NN search, relative retrieval confidence, dynamic span selection, and relaxed speculative decoding, NEST improves both validation perplexity and free-form generation quality on nine different tasks. Its

effectiveness can be further improved when combined with in-context retrieval augmentation. With these results, we demonstrate that NEST is capable of generating text grounded to real-world sources in low latency while maintaining fluency.

## 8 Broader Impact

The ability to copy real-world texts from existing data stores is useful for finding the source of the claim (credibility), preventing hallucination (factuality), as well as protecting copyright (risk management). It helps to resolve the dispute that often happens in AI tools by acknowledging the contents that are borrowed from existing human works (e.g., arts, books, and other creative content). Meanwhile, the information on the Internet is mixed and it is important to filter out false and sensitive information before directly injecting them into the generation.

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

# A Additional Implementation Details

**Two-stage $k$-NN search** For the first-stage passage search, we use a Faiss (Douze et al., 2024) dense index and Pyserini (Lin et al., 2021) BM25 index for efficient search. For the dense index, we first use DRAGON+ to encode each passage in the corpus into a single vector, and then use Faiss (index string "IVF65536,PQ256") to cluster the vectors into 65536 centroids and quantize them into 256 codes of 8 bits each. For the sparse index, we use the default hyper-parameters and the "optimize" option in Pyserini to reduce the index size. For the approximate nearest neighbor retrieval, we use nprobe= 4096. During passage search, we retrieve 4000 passages from each index and keep the similarity score for fusion. The fusion coefficient $\eta$ is determined by the relative confidence of dense and sparse retrievers similar to Equation (4). We set the dense coefficient $\eta_{\text{dense}} = 1 - \text{top-100}(s_{\text{dense}}(q, d)) / \max s_{\text{dense}}(q, d)$ and same for the sparse coefficient $\eta_{\text{sparse}}$. The final interpolation coef is $\eta = 0.7 * (1 - \eta_{\text{sparse}}) + 0.3 * \eta_{\text{dense}}$. The fusion score for each document $s(q, d) = \eta * s_{\text{dense}} + (1 - \eta) * s_{\text{sparse}}$. If a document is missing in either dense or sparse retrieval results, we set its score to the minimum similarity of the dense/sparse retrieval results. The first-stage search is done on RAM and CPUs with 32 threads. The final Wikipedia dense index size is about 8.96GB, and the sparse index size is about 3.48GB on disk.

For the second-stage token search, we use the LLM to encode the sequence and use the input to the final layer's feed-forward network after layer normalization as the key and query vectors following Khandelwal et al. (2020). We retrieve the top-1024 tokens using the squared $\ell_2$ distance and compute the non-parametric probability according to Equation (2).

**Rest of NEST** For relative retrieval confidence, we set $\alpha = 0.3, \tau = 0.1$ for all Wikipedia-based tasks and $\alpha = 0.2, \tau = 0.1$ for Pile of Law for all model sizes in Equation (4). For dynamic span selection, we set the n-gram length to be 64 and $\delta = 0.5$ for all model sizes and all tasks in Equation (6). For relaxed speculative decoding, we set $\gamma = 5e - 4$ for Pile of Law tasks for all model sizes in Equation (7). For Wikipedia-based tasks, we set $\gamma = 5e - 4$ for the 7B model, $\gamma = 5e - 3$ for the 13B model, and $\gamma = 5e - 2$ for the 70B model. For RA-NEST, all models use the same $\gamma = 5e - 1$ for all tasks except Pile of Law which still uses $\gamma = 5e - 4$ We observe that as the model gets stronger, using larger $\gamma$ which leads to more rejection, is more beneficial to generation quality. The complete NEST procedure is provided in Algorithm 1.

# B Evaluation Details and Hyper-parameter Tuning

**Datasets.** We sample subsets of WikiText-103, NQ, and Biography as dev sets for hyper-parameter tuning. We use the validation sets of WikiText-103 (CC BY-SA 3.0), NQ (Apache License 2.0), TriviaQA (Apache License 2.0), MedMCQA (MIT License), HotpotQA (Apache License 2.0), MMLU (MIT License), and Biography (MIT License) for validation. TruthfulQA (Apache License 2.0) only has the test set. We finally test all datasets shown in Table 1. For HotpotQA and MedMCQA, we do not have access to the test set and therefore the validation results are reported. For Biography, we use the labeled data that have human annotation as the validation and dev set, and the unlabeled data as the test set. In the original FACTSCORE paper, the authors use InstructGPT (Ouyang et al., 2022) to perform fact decomposition before verification. We hereby train our own decomposition model by further fine-tuning Llama-2 7B using publicly available datasets (Chen et al., 2022; Liu et al., 2023; Malaviya et al., 2024). For fact verification, we use the option `retrieval+llama+npm` to evaluate the decomposed atomic facts.

**Inference and prompts.** For language modelling and text completion, we use a context length of 128 tokens and a max generation length of 256 tokens. For the other tasks, we use 128 tokens as max generation length for question answering and 512 for fact verification. We remove all the in-context demonstrations from the prompt to test the zero-shot effectiveness of our model. We use greedy decoding in our experiments as the randomness in sampling can undermine factuality.

Regarding the prompts we use for evaluation, for MMLU, we compare the perplexity of each option concatenated with the question and select the one with the minimum perplexity.

For text completion, we use the following prompt "`[INST] Write an article.\n Article: [/INST]` {prefix}" where the `[INST]` is a format tag for Llama-2-Chat.

---

**Algorithm 1** NEST w/ Greedy Decoding

---

**Inputs:** Language model LM, hidden state encoder $f$, first-stage retriever $R$, corpus $\mathcal{C}$, input $x$.
▷ First-stage retrieval: Retrieve documents $d_1, d_2 \ldots, d_b$ from corpus $\mathcal{C}$
$d_1, d_2 \ldots, d_b \leftarrow R(x, \mathcal{C})$
▷ Second-stage retrieval: Construct token-level key-value memory
$(\mathcal{K}', \mathcal{V}') \leftarrow \varnothing$
**for** $i = 1$ **to** $b$ **do**
    $w_1^{d_i}, w_3^{d_i}, ..., w_m^{d_i} \leftarrow d_i$
    $h_1^{d_i}, h_3^{d_i}, ..., h_m^{d_i} \leftarrow f(d_i)$
    **for** $i = 1$ **to** $m - 1$ **do**
        $(\mathcal{K}', \mathcal{V}').\text{add}(h_j^{d_i}, w_{j+1}^{d_i})$
    **end for**
**end for**
▷ Generation
$y_{<t} \leftarrow x$
**for** $t = 1$ **to** $T$ **do**
    ▷ Compute query embedding
    $q_t \leftarrow f(y_{<t})[-1]$
    ▷ Token embeddings search, return top-$r$ scores and values
    $\pi \leftarrow (\mathcal{K}', \mathcal{V}').\text{search}(q_t, r)$
    $(s_1, v_1), (s_2, v_2), ..., (s_r, v_r) \leftarrow \pi$
    ▷ Compute non-parametric distribution
    $p_{k\text{-NN}}(w|y_{<t}) \leftarrow 0, \forall w \in \text{vocabulary}$
    **for** $i = 1$ **to** $r$ **do**
        $p_{k\text{-NN}}(w = v_i|y_{<t}) \leftarrow p_{k\text{-NN}}(w = v_i|y_{<t}) + \exp(\mu \cdot s_i) / \sum_{i=j}^{r} \exp(\mu \cdot s_j)$
    **end for**
    ▷ Confidence-based Interpolation
    $\lambda_t \leftarrow \text{sigmoid}((\frac{\min_i s_i}{\max_i s_i} - \alpha)/\tau)$
    $p_{\mathcal{M}}(w|y_{<t}) \leftarrow \lambda_t \cdot p_{\text{LM}}(w|y_{<t}) + (1 - \lambda_t) \cdot p_{k\text{-NN}}(w|y_{<t})$
    ▷ Dynamic span selection
    $w_t \leftarrow \underset{w}{\text{argmax}}\, p_{\mathcal{M}}(w|y_{<t})$
    $v_t \leftarrow \underset{v_i = w_t}{\text{argmax}}\, p_{k\text{-NN}}(w = v_i|y_{<t})$
    $v_{t:t+n} \leftarrow \mathcal{C}.\text{get-ngram}(v_t, n)$
    $y_t \leftarrow \begin{cases} w_t, & \text{if } \lambda_t > \delta; \\ v_{t:t+n}, & \text{otherwise.} \end{cases}$
    $n \leftarrow |y_t|$
    ▷ Relaxed Speculative Decoding
    **for** $i = 1$ **to** $n$ **do**
        $p_{\text{accept}}(w_t^{(i)}) \leftarrow \dfrac{p_{\mathcal{M}}(w=w_t^{(i)}|x, y_{<t}, w_t^{(1)}, w_t^{(2)}, ..., w_t^{(i-1)})}{\gamma \cdot \underset{w}{\max}\, p_{\mathcal{M}}(w|x, y_{<t}, w_t^{(1)}, w_t^{(2)}, ..., w_t^{(i-1)})}$
        Break if $p_{\text{accept}}(w_t^{(i)}) \leq 0.5$
    **end for**
    **if** $i < n$ and $n > 1$ **then**
        $w_t^{(i)} \leftarrow \underset{w}{\text{argmax}}\, p_{\mathcal{M}}(w|y_{<t}, w_t^{(1)}, w_t^{(2)}, ..., w_t^{(i-1)})$
    **end if**
    $y_{<t} \leftarrow \text{concatenate}(y_{<t}, w_t^{(1)}, w_t^{(2)}, ..., w_t^{(i)})$
**end for**
Return $y_{<t}$

---

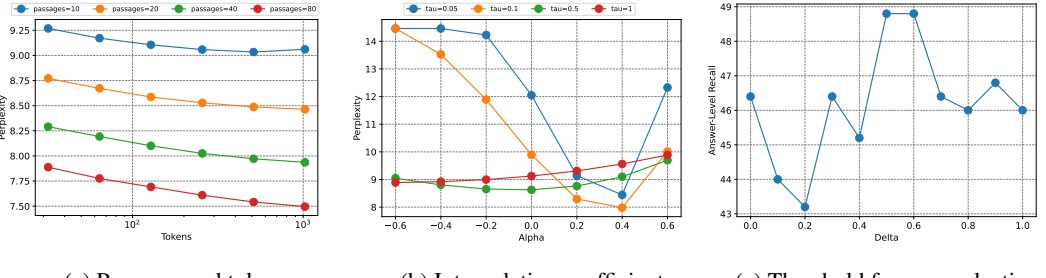

|  |  |  |
|---|---|---|
| (a) Passages and tokens. | (b) Interpolation coefficient. | (c) Threshold for span selection. |

Figure 3: Sensitivity analysis on WikiText-103 and NQ dev set for the NEST-7B model with the above hyper-parameters in the sub-figures.

For question-answering and fact-verification tasks, we use the following template: "`[INST]` Question: {question} Answer: `[/INST]`" where we format the input question in the bracket.

For the RA models, we use the prompt "`[INST]` Write an article with the background context as reference. Background: {retrieved passages}\n Article: `[/INST]` {prefix}" for text completion. For retrieval-augmented question-answering and fact-verification tasks, we use "`[INST]` Answer the question with the background context as reference. Background: {retrieved passages}\n Question: {question} Answer: `[/INST]`".

**Hyper-parameters and baselines.** For the base LM, we do not tune the hyper-parameters released with the original Llama-2-chat models. For the in-context retrieval augmented baseline, we select the top-3 retrieved passages. For $k$NN-LM, we follow Equation (3) and use an interpolation coefficient of 0.7 for Wikipedia-based tasks and 0.9 for Pile of Law. For NEST and RA-NEST, we first tune the hyper-parameters in Equation (4) on language modelling tasks using perplexity. We then fix those hyper-parameters and then tune the rest of the parameters in Equation (6) and (7) on generation tasks. All hyper-parameters in the above methods are tuned on the dev sets of WikiText-103, NQ, and Biography.

## C Analysis

The following analyses are performed on the validation set of WikiText-103, NQ, and Biography data with the Llama-7B-chat model.

### C.1 Sensitivity

**Number of retrieved passages and tokens** Khandelwal et al. (2020) show that increasing the size of the database and the number of tokens can improve the perplexity with proper hyper-parameter setting. We also verify whether our two-stage $k$-NN search and RRC approach follow the same trend. Figure 3a shows the validation perplexity on WikiText-103. For a fixed number of passages, the perplexity decreases as the number of tokens increases; for a fixed number of tokens, the perplexity decreases about $0.5 \sim 1.0$ as the number of passages doubles. However, as NEST needs to encode the retrieved passages on the fly, the latency also increases linearly w.r.t. the number of passages. Therefore, we set the passage number to be 40 and the token number to be 1024 in the main experiments.

**Interpolation coefficient** Figure 3b shows the sensitivity of the hyper-parameters $\alpha$ (offset) and $\tau$ (temperature) in Equation (4) on WikiText-103. When $\tau$ is big, $\lambda_t$ is close to a uniform distribution and therefore the offset $\alpha$ does not have a big impact on the perplexity. When $\tau$ is small, the impact of $\alpha$ is enlarged and the sweet spot is achieved around $\tau = 0.1$ and $\alpha = 0.4$.

**Threshold for dynamic span selection** Figure 3c shows how threshold $\delta$ in Equation (6) affects the generation on NQ. A bigger $\delta$ means selecting the span instead of a token more often. We can see that the answer-level recall on NQ first increases and then decreases as we increase the value of $\delta$, where the sweet spot is around $\delta = 0.5$.

| Models (7B) | Wiki./ROUGE-1 | NQ/ALR | Bio./FS |
|---|---|---|---|
| $k$NN-LM (two-stage) | 20.1 | 40.8 | 34.8 |
| + Relative Retrieval Confidence | 24.7 | 44.4 | 41.6 |
| + Dynamic Span selection | 24.5 | 44.6 | 41.6 |
| + Relaxed speculative decoding | 26.8 | 45.4 | 46.8 |

Table 3: Ablation study on the validation set of WikiText-103, NQ, and Biography. ROUGE-1 is reported for WikiText-103, ALR is reported for NQ, and FACTSCORE is reported for Biography.

## C.2  Ablation Study

Table 3 shows a progressive ablation of NEST on WikiText-103, NQ, Biography. As mentioned in Section 3.1, it is extremely expensive to encode billion-token corpus with billion-parameter models. Therefore, we directly start with the two-stage implementation of $k$NN-LM and gradually add the methods applied in NEST. As we can see, adding the RRC component gives the first effectiveness boost. The second dynamic span selection method does not seem to increase the effectiveness, yet it is crucial to give consistent attribution for consecutive spans and tokens. The last relaxed speculative decoding method further improves the final generation quality.

