# OpenReview forum: "Nearest Neighbor Speculative Decoding for LLM Generation and Attribution"
_NeurIPS.cc/2024/Conference — NeurIPS 2024 poster_

### Official Review · Reviewer_kPHS · 2024-07-09

**Soundness:** 3
**Presentation:** 3
**Contribution:** 3
**Rating:** 7
**Confidence:** 3

**Summary:**

The paper presents a new semi-parametric language modeling approach that can incorporate text spans from a datastore into LLM-based generation improving both quality and attribution of generated texts. They propose a two-step approach that requires constructing an on-the-fly token-level datastore based on a small number of retrieved passages from a passage-level datastore. The current token is generated from the mixture distribution between the base LLM and the retrieved token-level distribution interpolated using a relative retrieval confidence score capturing the uncertainty of the token retriever. Furthermore, the approach enables extending the generation from token to an n-gram span based on a speculative decoding that rejects or accepts the tokens in the continuations, thus enabling span-level generation, improving efficiency.

**Strengths:**

1. Extensive experiments across tasks and datasets shows the efficacy of the proposed approach over standard LLM decoding, KNN and Retrieval augmented incontext learning variants.
2. Ablations show how the relaxation factor used for speculative decoding can enable flexible-length n-gram continuations to be incorporated based on the domain and can provide a good tradeoff between accuracy and attribution.

**Weaknesses:**

1. Multiple-token nearest neighbor generation has been proposed in prior work but was not discussed or compared, see: [1] Chunk-based Nearest Neighbor Machine Translation (https://aclanthology.org/2022.emnlp-main.284/).

**Questions:**

See weaknesses.

**Limitations:**

Yes

---

> ### Author Rebuttal · Authors · 2024-08-02
>
> We thank the reviewer for recognizing the novelty and effectiveness of our work. The chunk retrieval and the neighbour selection process in the suggested paper are indeed related to NEST. For example, it is possible to improve the span selection process of NEST with the batch-beam-level neighbours in the paper which evaluate multiple chunk candidates simultaneously. We thank the reviewer again and will add a more detailed discussion of this paper in the related work section.

---

### Official Review · Reviewer_vMVQ · 2024-07-11

**Soundness:** 3
**Presentation:** 3
**Contribution:** 2
**Rating:** 5
**Confidence:** 4

**Summary:**

This paper:

* Introduces NEST, a semi-parametric language modeling approach that integrates real-world text spans into language model generations.
* Enhances generation quality and reduces latency by using token-level retrieval and speculative decoding.
* Outperforms conventional kNN-LM and competes well with in-context retrieval methods.
* Demonstrates significant improvements with various models.

**Strengths:**

* Introduces a novel semi-parametric language modeling technique for improved attribution and generation quality.
* Demonstrates a significant increase in speed and efficiency in language model generation.
* Outperforms conventional kNN-LM and shows competitive results against in-context retrieval methods across various knowledge-intensive tasks.
* Effective across a range of tasks including text completion, question answering, and factuality-aware generation, showcasing the method's adaptability to different content requirements.

**Weaknesses:**

* Performance heavily relies on the accuracy of the first-stage passage retrieval and second-stage token retrieval.
* The retrieval process may still be complex and resource-intensive for practical deployment.
* The gains in performance and efficiency are less pronounced in larger models.

**Questions:**

* How scalable is the NEST approach in real-world scenarios?
* What measures are taken to mitigate the impact of noise and errors introduced during the data retrieval stages?
* How does NEST address potential biases that may arise from the retrieved data?
* How well does NEST adapt to the evolving nature of language?

**Limitations:**

* The overall effectiveness of NEST is contingent upon the precision of both stages of text retrieval.
* The approach may struggle with ensuring that the retrieved content is contextually relevant and free from biases.

---

> ### Author Rebuttal · Authors · 2024-08-02
>
> We thank the reviewer for recognizing the novelty and effectiveness of NEST. Below, we address each weakness point raised:
> - *“Performance heavily relies on the accuracy of the first-stage passage retrieval”*:  We acknowledge the performance of NEST is impacted by the first-stage passage retrieval. However, all retrieval-augmented models have this limit, and our work focuses on improving 1) the integration of the retrieved content into the LLMs and 2), rather than further improving the first-stage retrieval accuracy.
> - *“Retrieval process may still be complex and resource-intensive for practical deployment”*: We politely disagree that the retrieval process is complex. Our two-stage retrieval system is designed to reduce the resources required for building a token-level datastore of the entire corpus, which is a significant advantage over other approaches. We have empirically demonstrated that our approach strikes a good balance between accuracy and efficiency.
> - *“The gains in performance and efficiency are less pronounced in larger models”*: We understand the reviewer's point that the gains in performance and efficiency are less pronounced in larger models. However, this is a natural consequence of the model's increased capacity to memorize nuanced facts. Nevertheless, retrieval augmentation remains meaningful in practice for long-tail knowledge, faster response, source attribution, and knowledge update. Moreover, our results show that the speedup is more prominent in larger models (the overall generation latency per query is dominated by the LM forwarding and rejection sampling process), making NEST a more efficient choice for large-scale applications.
>
> For the Questions:
> - **Scalability of NEST**: We believe that our experiments are representative of real-world scenarios, with the largest model having 70 billion parameters and the largest knowledge store having 33 million passages. Further engineering optimization can enable scaling to even larger models and knowledge stores.
> - **Measures to reduce noises and biases in retrieval**: The Confidence-Based Output Interpolation (sec 3.2) is our exact measure to mitigate the noise and biases in retrieval. Not incorporating the retrieval information at the input reduces the risk of biases and interpolating based on the token-retrieval confidence further filters out contradictory knowledge. This is also verified in the TruthfulQA experiments when adversarial queries are encountered, where NEST outperforms normal RAG methods by 0.3~0.73 points in Rouge-1. Please see line 219 - 223 for more details.
> - **Adapting to changes in language**: We interpret the reviewer's question as referring to the evolving nature of language, where new phrases and knowledge are added frequently. NEST's zero-shot combination of LLMs and knowledge sources enables it to adapt to these changes more easily. By tuning the hyperparameters on the development set, NEST can combine different LLMs with different knowledge sources, making it more flexible and adaptable to the evolving nature of language.

---

### Official Review · Reviewer_L2uT · 2024-07-12

**Soundness:** 4
**Presentation:** 3
**Contribution:** 2
**Rating:** 7
**Confidence:** 2

**Summary:**

This work presents Nearest Neighbor Speculative Decoding (NEST), a technique to better inject real-world text spans into the output of existing language models. NEST is a kNN-LM approach adding an initial passage retrieval step. During inference, NEST uses Relative Retrieval Confidence (RRC) for confidence-based interpolation, dynamically extends selected tokens to include text spans when confidence is high, and employs a relaxed speculative decoding process that accepts only highly probable token spans.

According to the evaluation conducted by the authors with Llama 2 chat, NEST significantly outperforms both the base LM and standard kNN-LM, in terms of speed and accuracy.

**Strengths:**

- well-written: The paper is clear and easy to understand. Figure 1 itself is enough to understand what the paper presents
- well-motivated: The paper explains the problem that it tries to solve and provides the motivations for why it needs to be solved
- Extensive experiments demonstrating that the proposed approach works

**Weaknesses:**

- It’s unclear to me what are the most important contributions of this work. What is really new and has the most impact on the results? Even with the help of the related work section I cannot answer this question, hence my low confidence score.

**Questions:**

It’s unclear to me what are the most important contributions of this work. What is really new and has the most impact on the results? Even with the help of the related work section I cannot answer this question, hence my low confidence score

**Limitations:**

The limitations are pointed out and sufficiently discussed.

---

> ### Author Rebuttal · Authors · 2024-08-02
>
> We appreciate the reviewer's feedback and would like to clarify the main contributions of NEST:
> - **Source Attribution**: By injecting retrieved segments into the generation of LLM, NEST provides direct attribution for the generation at a span level, enabling users to verify the reliability of the generation. Methods based on prompting/finetuning to provide citations can still make mistakes, while for NEST, given the same set of retrieved documents, the attribution of the claim is always correct since the segments are directly taken from the source.
> - **Copyright Protection**: Using LLMs for content generation is controversial because these models were probably trained on copyrighted content, which may infringe on the rights of data producers. Storing the proprietary data separately in a data store avoids memorizing high-risk knowledge; providing direction attribution to each claim avoids the legal risk of using licenced data without acknowledgement.
> - **Factuality**: Grounding the generation using spans taken from a verified knowledge source can improve the factuality of LLM generation. Unlike normal RAG settings where the retrieved evidence is injected into the prompt, NEST can choose not to use the retrieved results during interpolation. This is also showcased in adversarial question-answering tasks such as TruthfulQA, where RAG can easily be misled by the retrieved “evidence” that is seemingly related but incorrect, while NEST can avoid such misleading knowledge by not interpolating with them at the output. Please see line 219 - 223 for more details.
>
> Moreover, we would like to point out some technical improvements NEST made in LLM and retrieval-augmented generation:
> - **Dynamic Interpolation**: Retrieval is not always required for tasks such as creative writing, while RAG and baseline kNN-LM methods both used a fixed scheme to incorporate retrieved “evidence”, which might hurt the generation in such cases. Other methods that can adaptively choose to combine the retrieved knowledge such as CoG or Retro require further finetuning, which couples the LLM with the knowledge base and might not be able to transfer to new databases. On the other hand, NEST can dynamically adjust the interpolation between LLMs and retrievers in a zero-shot fashion, making it easy for developers to update the LLM and knowledge base.
> - **Generation Efficiency**: Injecting the retrieved segment into LLM generation also improves the generation speed thanks to speculative decoding. We did not optimize for this aspect but using a faster and better retriever should further improve the generation latency.
> - **Attribution-Fluency Tradeoff**: It might not be obvious in the paper but injecting segments into LLM generation (attribution) is risky as it might introduce artifacts during transition such as repetition or grammatical errors (fluency). This is hard to evaluate but has a major impact in real application. Our proposed dynamic span selection and relaxed speculative decoding method provide a solution for attribution-fluency trade-off by tuning the selection and rejection hyperparameters in Equation (6) and (7).
>
> In summary, NEST not only solves the technical challenges in previous generation methods, such as when to combine with retrieval and attribution-fluency tradeoff but also has real impacts on applications such as copyright protection and providing accurate attribution for verification. We again thank the reviewer for pointing this out and we will elaborate on the contribution and novelty of NEST in the paper.

---

> ### Author Response · Authors · 2024-08-13
> **Respectfully Asking to Reconsider Your Rating**
>
> Thank you again for your feedback.
>
> We would like to follow up on our rebuttal to ensure that we have thoroughly addressed your initial concerns, and to respectfully ask for a reconsideration of the overall rating. Specifically, we would like to provide additional clarification on the two areas of concerns raised.
>
> (1) **Novelty:** Our proposed approach, NEST, primarily consists of two novel techniques: (1) a confidence-adjusted, token-level retrieval score to extract text spans from real-world corpora as draft inputs, and (2) a relaxed-speculative decoding procedure to seamlessly integrate these drafts into the LLM generation process, rejecting the uncertain suffixes. We demonstrated empirically that this approach allows attribution of the generated text directly to the source, and yields significant improvements in both generation speed and quality compared to the base LLMs and kNN-LM with two-stage retrieval. kNN-LM with two-stage retrieval is a stronger baseline compared to the standard kNN-LM. Reviewers nNiG, vMVQ, and kPHS have also recognized the novelty and strengths of our approach.
>
> (2) **Which technique had the most impact on performance:** We included an ablation study in the appendix of our submission (**Appendix C.2, Table 3**). The results show that both adjusting the interpolation score based on token-level retrieval confidence (+4.6 ROUGE-1 for WikiText-103 and +6.8 FactScore on Biography) and performing token rejection using relaxed speculative decoding (+2.3 ROUGE-1 for WikiText-103 and +5.2 FactScore on Biography) significantly contribute to the performance improvements.
>
> We plan to move the ablation study to the main text to provide clearer insights for future readers with additional main content space.
> | Models (7B)                     | Wiki./ROUGE-1 | NQ/ALR | Bio./FS |
> |---------------------------------|---------------|--------|---------|
> | kNN-LM (two-stage)              | 20.1          | 40.8   | 34.8    |
> | + Relative Retrieval Confidence | 24.7          | 44.4   | 41.6    |
> | + Dynamic Span selection        | 24.5          | 44.6   | 41.6    |
> | + Relaxed speculative decoding  | 26.8          | 45.4   | 46.8    |

---

### Official Review · Reviewer_nNiG · 2024-07-13

**Soundness:** 2
**Presentation:** 2
**Contribution:** 2
**Rating:** 5
**Confidence:** 4

**Summary:**

The paper introduces NEST, a novel semi-parametric language modeling approach that enhances the generation quality and attribution of Large Language Models (LLMs) by incorporating real-world text spans. NEST employs a two-stage k-NN search and speculative decoding, achieving improved performance and reduced inference time.

**Strengths:**

The two-stage k-NN search is a smart optimization that balances search accuracy and efficiency.

The paper demonstrates significant improvements in generation quality and speed, offering a competitive edge over traditional methods.

The approach of providing direct attribution to sources is valuable for enhancing the reliability of LLMs.

**Weaknesses:**

The paper could benefit from a more detailed comparison with state-of-the-art methods.

The generalizability of NEST across different domains and languages needs further exploration.

The potential impact of NEST on the in-context learning ability of LLMs is not thoroughly discussed.

The paper lacks a comprehensive analysis of error rates and statistical significance.

**Questions:**

How does NEST compare with other advanced language models in terms of handling long-tail knowledge?

What are the implications of using NEST for models pre-trained with different objectives or datasets?

Could the authors elaborate on the impact of NEST on the diversity and creativity of LLM generations?

On page 2, the authors mention a 1.8× speedup; could they specify if this is consistent across different model sizes?

How does NEST handle potential biases in the retrieved text spans from the corpus?

Can the authors provide more insight into the decision process behind the choice of hyperparameters?

Is there a risk of overfitting the corpus used for training the key-value datastore?

---

> ### Author Rebuttal · Authors · 2024-08-02
>
> We thank the reviewer for recognizing the contributions of NEST. Below, we address each weakness point raised:
> - **Comparison with SOTA models**: We appreciate the reviewer's comment on comparing NEST with state-of-the-art (SOTA) models. To clarify, NEST is an algorithm that can be used to combine any SOTA LLMs with a desired data store to improve generation and attribution. While the retrieval-augmented generation (RAG) baseline outperforms NEST on several tasks, we highlight that NEST can be further combined with RA to achieve better performance (Table 1, last row). This flexibility is a key strength of our approach.
> - **Generalization of NEST across different domains**: We evaluated NEST across multiple domains, including general, legal, and medical fields, as well as a diverse set of subjects covered by MMLU (sec 4.4). Our results show that NEST consistently improves performance across these domains. While our current focus is mainly on English due to the availability of robust open-sourced multi-lingual LLMs, we believe that NEST can be applied to other languages with minimal modifications.
> - **In-context learning ability**: As mentioned in the Limitation section, NEST may not improve or even slightly hurt in-context learning since the demonstrations in the prompts are unlikely to appear in the natural text predominantly present in the retrieval data store. To quantify the impact, we conducted an experiment using Llama-2 (7B and 13B, not instruction-tuned) with 5-shot demonstrations on the Natural Questions dev set. Our results show that NEST does not improve performance in this setting. We will include these additional results in future versions of the paper.
>
> | Models |	                EM |	   F1 |	ALR|
> | -------- | ------- | ------- | ------- |
> | Llama-2-7B |	24.05	| 33.53	| 27.10 |
> | +NEST	    |          23.50 |	32.87 |	27.06 |
> | Llama-2-13B |	30.34	| 41.17	| 33.76 |
> | +NEST	  |            30.07	| 40.77	| 34.10 |
>
> For the Questions:
> - **Handling long-tail knowledge**: Semi-parametric LMs such as NEST have stronger capabilities in memorizing long-tail facts because of the direct access to a non-parametric data store. Our experiment on MedMCQA, which involves medical questions that are often "long-tail," shows that NEST outperforms the base LM model by 0.7~1 point in answer recall (Table 1).
> - **Different base LMs**: NEST is primarily designed for auto-regressive language models to improve text generation. Within this scope, NEST is designed to be flexible and can be combined with different LLMs and data stores in zero-shot, and we used Llama-2-chat as an example in the paper. We notice the notion of “models **pre-trained with different objectives or datasets**” can be ambiguous, and we’re happy to further clarify if we misunderstood the reviewer’s question.
> - **Diversity and creativity**: While NEST primarily focuses on improving factuality and attribution, we believe that our approach can be extended to incorporate diversity and creativity in future work. The Confidence-Based Output Interpolation mechanism we propose (sec 3.2) can determine whether to interpolate the LM prediction with the retrieved evidence. For creative generation scenarios, NEST will primarily rely on the base LLM for generation.
> - **Efficiency across different model sizes**: The speedup is not the same across models of different sizes. The overall generation latency per query is dominated by the LM forward pass and the rejection sampling process, while the other parts (retrieval and index building) stay almost constant. Therefore, the speedup is more obvious for larger LLMs and less prominent for smaller models, which is consistent with other speculative decoding methods.
> - **Biases in retrieval**: The Confidence-Based Output Interpolation mitigates the noise and biases in retrieval. By using the confidence of the retrieval, the LLM can adjust the frequency to interpolate with the retrieval distribution. This is verified in the TruthfulQA experiments, where adversarial queries are prevalent. NEST outperforms normal RAG methods by 0.3~0.73 points in Rouge-1 (line 219 - 223).
> - **Choice of hyperparameters**: We provide a detailed hyperparameter tuning scheme in Appendix A and B. The parameters $\alpha$ and $\tau$ in Equation (4) reflect the prior of how trustworthy the retriever is, and using larger $\alpha$ means trusting the retriever more.
> For $\delta$ in Equation (5), we find 0.5 works best across all models and tasks, which is reasonable as the interpolation coefficient is well-calibrated in Equation (4).
> For $\gamma$ in Equation (6), it determines how often the LM rejects a segment retrieved from the corpus. A smaller $\gamma$ means less rejection. We find that using a larger $\gamma$ for a stronger LM (more rejection) tends to work better.
> - **Risk of corpus overfitting**: We would like to clarify that our LLMs and retriever are not fine-tuned on the corpus we used for retrieval. On the other hand, we tuned the hyperparameters in Equation (4), (6), and (7) for each retrieval corpus, assuming that the information distribution and quality of the data were unknown until tested. We recommend practitioners adopting the NEST approach also tune the hyperparameters on their retrieval corpus. This is cheap to complete and may significantly boost downstream performance.

---

> ### Comment · Reviewer_nNiG · 2024-08-13
>
> I have read the author's response and other reviews, and I will keep my rating unchanged.

---

> ### Author Response · Authors · 2024-08-13
>
> Thank you again for your feedback. We're happy to address any remaining concerns you may have and to provide further clarifications to ensure a thorough understanding of our research.

---

### Decision · Program_Chairs · 2024-09-25

**Decision:**

Accept (poster)

**Comment:**

The paper presents a competitive technique to incorporate real-world text spans of arbitrary length into LM generations, thus enhancing also attribution.
All major concerns from reviewers appear resolved during rebuttal.